# Development of a Portable Residual Chlorine Detection Device with a Combination of Microfluidic Chips and LS-BP Algorithm to Achieve Accurate Detection of Residual Chlorine in Water

**DOI:** 10.3390/mi15081045

**Published:** 2024-08-18

**Authors:** Tongfei Wang, Jiping Niu, Haoran Pang, Xiaoyu Meng, Ruqian Sun, Jiaqing Xie

**Affiliations:** 1College of Mechanical and Electronic Engineering, Northwest A&F University, Yangling 712100, China; 1665831379@nwafu.edu.cn (T.W.); 18729071855@nwafu.edu.cn (H.P.); mxy0917@nwafu.edu.cn (X.M.); sunruqian@nwafu.edu.cn (R.S.); xiejq@nwafu.edu.cn (J.X.); 2College of Water Resources and Architectural Engineering, Northwest A&F University, Yangling 712100, China

**Keywords:** residual chlorine, microfluidic chip, LS-BP algorithm, deep learning algorithms

## Abstract

Chlorine is widely used for sterilization and disinfection of water, but the presence of excess residual chlorine in water poses a substantial threat to human health. At present, there is no portable device which can achieve accurate, rapid, low-cost, and convenient detection of residual chlorine in water. Therefore, it is necessary to develop a device that can perform accurate, rapid, low-cost, and convenient detection of residual chlorine in water. In this study, a portable residual chlorine detection device was developed. A microfluidic chip was studied to achieve efficient mixing of two-phase flow. This microfluidic chip was used for rapid mixing of reagents in the portable residual chlorine detection device, reducing the consumption of reagents, detection time, and device volume. A deep learning algorithm was proposed for predicting residual chlorine concentration in water, achieving precise detection. Firstly, the microfluidic chip structure for detecting mixed reagents was optimized, and the microfluidic chip was fabricated by a 3D-printing method. Secondly, a deep learning (LS-BP) algorithm was constructed and proposed for predicting residual chlorine concentration in water, which can realize dual-channel signal reading. Thirdly, the corresponding portable residual chlorine detection device was developed, and the detection device was compared with residual chlorine detection devices and methods in other studies. The comparison results indicate that the portable residual chlorine detection device has high detection accuracy, fast detection speed, low cost, and good convenience. The excellent performance of the portable residual chlorine detection device makes it suitable for detecting residual chlorine in drinking water, swimming pool water, aquaculture and other fields.

## 1. Introduction

Water quality has become a global concern in recent years [1]. Chlorine is widely used in the treatment of tap water due to its good sterilization effect and rapid sterilization speed [2,3,4]. The total concentration of dissolved chlorine (${\mathrm{Cl}}_{2}$), hypochlorous acid (HClO), and hypochlorite ions (${\mathrm{ClO}}^{-}$) is defined as free residual chlorine [5,6,7]. Residual chlorine in tap water needs to be at a certain concentration to ensure that the tap water meets the usage standards [8]. According to the standards of the World Health Organization, the recommended concentration of free chlorine in domestic water is between 2.0 mg·L^−1^ and 5.0 mg·L^−1^ [9]. If a certain concentration of residual chlorine cannot be maintained, it may have catastrophic consequences, such as those that occurred during the Walkerton Tragedy (Canada), when an E. coli outbreak caused seven deaths and 2300 infections [10,11]. Residual chlorine can react chemically with residual organic matter in water to form a series of chlorinated hydrocarbons such as chloroform, bromoform, and bromodichloromethane [12,13,14]. These substances are carcinogenic to humans [15]. Therefore, there is a great demand for the detection of residual chlorine in water. The electrochemical method [16,17], chromatography [18,19], and mass spectrometry [20] are traditional residual chlorine detection methods. Electrochemical method has accurate detection results and a short response time, but the detection results are affected by factors such as electrode aging, and the size of the detection device is large, and thus it cannot meet the needs of convenient portability and on-site detection [21]. Chromatography, mass spectrometry, and chemiluminescence require complex pretreatment processes, high requirements for detection equipment, and high detection costs, and are not suitable for on-site detection and daily use [22]. At present, there is no portable device that can achieve accurate, rapid, low-cost, and convenient detection of residual chlorine in water.

Yen et al. studied a portable nanohybrid paper-based chemiresistive sensor for detecting residual chlorine in water [23]. However, this method has a longer detection time and lower integration of the detection device, which provides some room for improvement. Dou et al. studied a smartphone-based colorimetric device with improved sensitivity and accuracy for field analysis of residual chlorine in water samples [24]. The detection device has good reliability and good portability, but with a single signal reading, the detection results may be affected by other factors, which can be further improved. Yin et al. studied a batch microfabrication of a self-cleaning, ultradurable electrochemical sensor employing a BDD film for the online monitoring of residual chlorine in tap water [25]. Zhou et al. studied membrane-based portable colorimetric gaseous chlorine sensing probe [26]. Kodera et al. studied electrochemical detection of residual chlorine using Ni metal nanoparticles combined with multilayered graphene nanoshells [27]. These devices have good detection accuracy, but they may be affected by electrode oxidation, leading to detection issues. Charles conducted a study using six deep learning and nine machine learning techniques to predict residual chlorine [28], and the results showed that using deep learning and machine learning techniques can improve the accuracy of residual chlorine prediction. Mu et al. studied a microfluidic system for residual chlorine detection based on spectrophotometry [29] and achieved rapid and low-cost detection of residual chlorine, but there was still room for improvement in detection accuracy and reliability. Therefore, there is an urgent need to develop a portable residual chlorine detection device which can accurately, rapidly, and conveniently detect residual chlorine in water at low cost. 

In recent years, microfluidic systems have attracted widespread research attention in fields such as analytical chemistry and rapid detection, as they provide a miniaturized platform for traditional analytical techniques. Compared to traditional methods, microfluidic systems allow for faster and lower-cost analysis using fewer samples and reagents [30,31]. Microfluidic chips can be used to build portable detection devices for rapid detection of harmful substances in water bodies. The rapid detection device undoubtedly attracts the favor of users. Its operation is simple and it can be used on-site and simultaneously, while microfluidic chips can meet the needs of rapid detection [32]. Microfluidic chips can miniaturize and integrate chemical operations such as sample preparation, reaction, and detection into a small chip of only a few square centimeters in size, and microfluidic chips have the advantages of rapid and convenient detection [33,34]. Pang et al. developed an organophosphorus pesticide detection device based on microfluidic chips, which reduced the detection time for organic phosphorus pesticides from over 10 min to 1 min [35]. Hao et al. developed a microfluidic paper-based chip for the simultaneous rapid detection of fenbuconazole and dimethomorph. This chip enables fast quantitative detection of mixed pesticide residues within 5 min [36]. Ricardo A.G.de Oliveira et al. proved that microfluidic chips can reduce the detection time and consumption of detection reagents [37]. Based on the above analysis, microfluidic chips have shown great potential in the detection of harmful substances in solutions.

At present, detection devices usually adopt a single-channel signal reading method. Although this method is easy to operate, its detection accuracy is susceptible to external interference. The dual-channel signal reading method can effectively reduce the interference generated by the outside world and has better detection accuracy [38]. Due to the potential impact of other impurities in the solution on the single-channel signal reading method [39], the dual-channel signal reading method has higher analysis efficiency, analysis performance, and higher detection accuracy, and it expands the linear range of detection [40]. Zhao et al. developed a dual-channel signal reading device for the detection of carbamate pesticides, and the device exhibited high detection precision with a relative standard deviation of less than 4.5% [41]. Yan et al. developed an organic phosphorus detection sensor based on single-enzyme inhibition with dual-channel signal reading. The sensor demonstrated a spiked recovery rate between 87.0% and 118.0%, with a standard deviation of less than 5.1% [42]. Liu et al. developed a dual-channel signal reading device for the detection of organic phosphorus and carbamate pesticides. The device achieved remarkably low minimum detection concentrations of 0.1 μg·L^−1^, 0.3 μg·L^−1^, and 1 μg·L^−1^ for azinphos-methyl, malathion, and phosphates, respectively. These low minimum detection concentrations were much lower than the maximum residue limits reported in the EU pesticide database [43]. The above research indicates that dual-channel signal reading method can effectively reduce the impact of external interference on detection results and can also improve detection accuracy and precision.

Deep learning algorithms have received widespread attention in recent years and have been widely applied in various fields. They have also shown great potential in the field of food inspection [44]. The use of deep learning for detecting harmful substances in food can effectively improve detection accuracy. Ye et al. proposed a machine learning-based hyperspectral imaging technique for detecting pesticide residues in grapes. The accuracy of the detection results for pesticide residues exceeded 95% [45]. Wang et al. proposed a Mind Evolutionary Algorithm (MEA) derived from a genetic algorithm for detecting residual levels of two carbamate pesticides on tomatoes. The relative error and average recovery rates for the detection of carbamate pesticide residues on tomatoes were reported as 1.325%, 2.375%, 98.94%, and 99.25%, respectively [46]. Sun et al. employed a Convolutional Neural Network (CNN) algorithm for detecting pesticide residues on lettuce leaves. The detection results achieved a root mean square error (RMSE) of 0.134 mg·L^−1^, meeting the accuracy requirements outlined in the Chinese national standards [47]. The above analysis indicates that deep learning algorithms can improve the detection accuracy of harmful substances in food.

Existing studies have shown that microfluidic chips, deep learning algorithms, and dual-channel signal reading have significant potential with regard to accurate, rapid, low-cost and convenient detection of harmful substances in food. Based on the above analysis, a portable residual chlorine detection device with a combination of microfluidic chips and deep learning algorithms was developed. Firstly, the parameters of the microfluidic chip structure were modeled and optimized, and the microfluidic chip was prepared by 3D printing method. This microfluidic chip can achieve rapid mixing of test solution and detection solution in the residual chlorine detection device, reducing detection time, device volume, and the consumption of detection reagents. Secondly, a deep learning (LS-BP) algorithm which can realize dual-channel signal reading was constructed and proposed to predict residual chlorine in water. The LS-BP algorithm is based on a least squares method and a BP neural network. This algorithm first uses the least squares method to obtain prediction residuals, then trains a BP neural network to obtain a compensation function, and finally obtains a prediction function with smaller prediction errors. The construction and proposal of this algorithm has positive significance for improving the detection accuracy of the detection device. Thirdly, the portable residual chlorine detection device was developed, and the device was compared with residual chlorine detection devices and methods in other studies. The comparison results showed that this portable residual chlorine detection device offers high detection accuracy, fast detection speed, low cost, and excellent convenience. The portable residual chlorine detection device can detect residual chlorine in daily drinking water, swimming pools, aquaculture, and other related fields, which is of great significance in safeguarding public health. Furthermore, this detection device offers both theoretical insights and empirical evidence for detecting harmful substances in water.

## 2. Experiments

### 2.1. Development of the Portable Residual Chlorine Detection Device

The development of the portable residual chlorine detection device was based on the design of microfluidic chips, the proposal of deep learning algorithms, circuit design, and the implementation of dual signal reading. The design and structural optimization of microfluidic chips can achieve rapid mixing of test solution and detection solution, reducing the consumption of detection reagents, detection time and device volume. The proposal of deep learning algorithms can improve detection accuracy. The design of hardware circuits can improve the integration and portability of the device. The dual signal reading can improve the anti-interference and accuracy of detection results.

The working principle of the portable residual chlorine detection device is shown in Figure 1. The STM32 microcontroller outputs PWM signals to control micro peristaltic pumps, achieving control of the liquid flow rate in microfluidic chip. STM32 microcontroller controls LED constant light source and indirectly measures the absorbance of liquid in microfluidic chip through photoresistors, which is collected in the form of voltage. The collected voltages are processed by STM32 microcontroller to obtain the detection results, and STM32 microcontroller displays the detection results through an OLED screen and HC-05 Bluetooth module.

An integrated circuit was designed to reduce circuit volume, and the designed integrated circuit is shown in Figure 2. The main electronic components of the integrated circuit are as follows: the integrated circuit control is controlled by a microcontroller, and the specific model is STM32F103C8T6. The integrated circuit USB to serial port is controlled by a serial port chip, and the specific model is CH340G. The integrated circuit power supply is provided through a USB interface, and the specific model is USB-M. The reception of integrated circuit optical signals is achieved by photoresistors, and the specific model is RG5528. The restart of the integrated circuit is controlled by the reset button, and the specific model is SW-PB. The voltage regulation of integrated circuits is controlled by a voltage regulator chip, and the specific model is ME6211C33M5G-N. The clock signal of the integrated circuit is provided by the crystal oscillator, and the specific models are HC-49S, OSC 3215-2P, 5032.

The length, width, and height of the portable residual chlorine detection device is 110 mm long, 110 mm wide, and 60 mm high, respectively, which can meet the need of portability. The composition of the portable residual chlorine detection device is shown in Figure 3. The portable residual chlorine detection device consists of an integrated circuit, LED constant light source, a constant voltage power supply, filters, microfluidic chips, an OLED screen, a HC-05 Bluetooth module, and micro peristaltic pumps. To ensure that the detection device has a good sealing performance, the microfluidic chips in the detection device were encapsulated with nano paste, the microfluidic chips were affixed to the joint base with 502 adhesive, and the connection between the joint base and the joint utilized threading to prevent any liquid leakage. And during multiple experiments using the device, no liquid leakage was observed in the microfluidic chips, demonstrating the excellent sealing performance.

The detection process is as follows: firstly, the integrated circuit controls the micro peristaltic pumps to pump the liquid to be detected and detection liquid into the microfluidic chips at a predetermined flow rate and controls the LED constant light source to illuminate observation ports through the filters. Secondly, the photoresistors on the integrated circuit receive the light signals from observation ports and transmit the light signals to the integrated circuit in the form of voltage. Thirdly, the internal program of the STM32 microcontroller processes the collected voltage signals and obtains the detection results. The detection results are presented on the OLED screen and transmitted to the phone through the HC-05 Bluetooth module.

### 2.2. Methods

Usually, limit of detection (LOD) [48,49] and relative standard deviation (RSD) [50] are two important indicators used to evaluate the detection reliability of detection devices. The detection limit is used to indicate the minimum concentration or quantity of the component to be tested that can be detected from the tested sample under certain conditions [51], and the relative standard deviation is used to indicate the degree of dispersion of the detection results, reflecting the accuracy of the detection results [52].

The calculation formula for limit of detection defined by IUPAC is as follows [53]:(1)$$X_{L}={\bar{X}}_{b}+KS_{b}$$
where $X_{L}$ is the limit of detection, ${\bar{X}}_{b}$ is the blank mean value, K is a constant related to confidence which is recommended by IUPAC to take 3.3, $S_{b}$ is the standard deviation of blank value, and ${\bar{X}}_{b}$ and $S_{b}$ need to be determined many times. Due to measurement errors, the low concentration level may not follow the normal distribution, and the number of blank measurements is limited, and the corresponding confidence level of K taking 3 is about 90% [53]. Usually, blank samples are measured 10 times [54].

The calculation formula of relative standard deviation is as follows:(2)$$RSD=\frac{S_{D}}{{\bar{X}}_{D}}\times 100\%$$
where *RSD* is the relative standard deviation, $S_{D}$ is the standard deviation of multiple measurements of a certain concentration, and ${\bar{X}}_{D}$ is the mean value of multiple measurements of a certain concentration. Usually, samples of different concentrations are measured three times to reduce measurement errors [55].

The portable residual chlorine detection device was used to detect residual chlorine standard solutions ranging from 1 mg·L^−1^ to 10 mg·L^−1^, with three measurements taken every 1 mg·L^−1^, and the device was used to detect the blank samples 10 times to obtain relevant data and calculate LOD and RSD.

### 2.3. Fluid Simulation Mechanics Model

The diffusion coefficient *D* can be defined as follows:(3)$$D=\frac{kT}{6\pi \mu r}$$
where *k* is the Boltzmann constant, *T* is the absolute temperature, *μ* is the dynamic viscosity, and *r* is the molecular radius. The diffusion coefficient is inversely proportional to the dynamic viscosity of the solution at a certain temperature.

The section structure of a mixing channel of the fabricated microfluidic chip is rectangular, and the expression of Reynolds number can be expressed as follows:(4)$$R_{e}=\frac{4\rho Av}{p\mu }$$
where A is the interface area, ρ is the liquid density, p is the wetting perimeter, and v is the flow rate.

The Navier–Stokes equation, which describes the behavior of incompressible fluid, is used to simulate the mass and momentum transfer of fluid. The expression is as follows:(5)$$\frac{\partial }{\partial x_{j}}\left(\rho u_{j}\right)=0$$
(6)$$\frac{\partial }{\partial x_{i}}\left(\rho u_{i}u_{j}\right)-\frac{\partial P}{\partial x_{i}}+\frac{\partial \tau_{ij}}{\partial x_{i}}$$
where ρ and $u_{j}$ are the density vector and velocity vector, respectively, u is the velocity vector of the fluid, *P* is the pressure on the fluid, and $\tau_{ij}$ is the stress tensor, respectively.

The mass flux is given by diffusion and convection, and the obtained mass balance is:(7)$$\nabla \cdot \left(-D\nabla c+c\vec{u}\right)=0$$
where *c* represents the concentration.

### 2.4. Parameterized Design of Microfluidic Chip

The liquid mixing efficiency of the fluid can improve the detection accuracy and reduce consumption of reagents, so it is necessary to design a microfluidic chip with good liquid mixing efficiency. The liquid flow rate in the channel is very small, and the corresponding Reynolds number is also very low, and the solute exchange between the fluids is mainly achieved by molecular diffusion. Therefore, it is necessary to apply channel structure bending and cross-sectional size changes to the design of microfluidic chips to enhance mass transfer to improve liquid mixing efficiency.

Figure 4 is the schematic diagram of the mixing channel structure for microfluidic chips. The microfluidic chip is divided into two inlets, one outlet, a mixing channel, and an observation port. The mixing channel was optimized through a parameterized design method, and the parameterized model was expanded according to the following formula:(8)Y=Asin(ωX)
where parameter *α* is the channel width, parameter *β* is *2π/ω*, *ω* is the channel angular frequency, *A* is the channel amplitude, and *γ* is double *A*. 

The factors affecting the liquid mixing efficiency of the microfluidic chip are the channel amplitude *A*, the channel width *α*, and the channel angular frequency *ω*. In order to explore the influence of these three factors on the qualitative liquid mixing efficiency, the fluid dynamics simulations of the liquid mixing in the microfluidic chip under different parameters were carried out. The flow characteristics of the microfluidic chip were studied under two-dimensional, steady-state conditions using the physical field of laminar, dilute species transfer. The governing equations were the continuity equation, incompressible fluid equation, Navier–Stokes equation, mass balance equation and diffusion–convection equation. The parameter optimization intervals are channel amplitude *A* from 0.5 mm to 2 mm, channel width *α* from 0.5 mm to 2 mm, and channel angular frequency *ω* from 0.5 rad·s^−1^ to 2 rad·s^−1^. According to the simulation results, the liquid mixing efficiency of each cross-section of the microfluidic chip under different parameters and a set of parameters with the best liquid mixing efficiency were obtained.

The two methods for detecting residual chlorine were N, the N-diethyl-p-phenylenediamine detection method (DPD colorimetric method) and the o-tolidine detection method (OTO colorimetric method). O-tolidine had the largest molecular size among the substances involved in these two colorimetric methods, and its diffusion rate was the smallest. Therefore, the standard deviation of o-tolidine concentration on different cross-sections of the microfluidic chip was used to measure the uniformity of fluid distribution and the liquid mixing efficiency.

The mixing index was introduced to represent the liquid mixing efficiency, and it referred to the standard deviation of the maximum and minimum diffusion concentration of the mixture on different cross-sections of the microfluidic chip. The calculation formula is
(9)$$M=\sqrt{\frac{1}{N}\sum_{i=1}^{N}{\left(C_{i}-\bar{C}\right)}^{2}}$$
where $C_{i}$ is the maximum and the minimum concentration of o-toluidine on the statistical cross-section, $\bar{C}$ is the average value of the maximum and minimum concentration of o-toluidine on the statistical cross-section, and *N* is the selected concentration number of the statistical cross-section. Mixing is more uniform when the difference between the maximum and the minimum concentration of the channel cross-section is smaller.

A coordinate was established on the microfluidic chip, with sections spaced 5 mm apart horizontally, and the liquid mixing starting point of the microfluidic chip was selected as the first section. The liquid injection rate was 2 × 10^−9^ m^3^·s^−1^. The initial o-toluidine concentration at the inlets of the microfluidic chip was selected as 1 mol·L^−1^ and 0 mol·L^−1^ to better reflect the liquid mixing efficiency.

### 2.5. The Principle of Dual-Channel Signal Reading Method

The signals of the dual-channel signal reading were the absorbances of color development solution of the DPD colorimetric method and the OTO colorimetric method. The principles of these two colorimetric methods are shown in Figure 5 [56]. The principle of the DPD colorimetric method is that N, N-diethyl-p-phenylenediamine reacts with residual chlorine under acidic conditions and forms a red compound. The principle of the OTO colorimetric method is based on the redox reaction between o-toluidine and residual chlorine, which forms yellow dihydrochloric acid quinone o-toluidine, and the color reaction is yellow. The color result of the DPD colorimetric method is red, and the color result of the OTO colorimetric method is yellow. The red solution has the maximum absorbance at 490–510 nm wavelength, and the yellow solution has the maximum absorbance at 440–460 nm wavelength [57]. Light at 440–460 nm and 490–510 nm wavelengths was used to irradiate the solutions after color development, respectively.

The detection of residual chlorine in water through the DPD colorimetric method or the OTO colorimetric method may be affected by impurities in the solution. The impurities probably have a strong absorption capacity for the light of certain wavelengths. This may cause fluctuations in the absorbance of the solution and lead to inaccurate detection results and deviation of the detection results.

When the DPD colorimetric method and the OTO colorimetric method are used simultaneously to detect residual chlorine, the wavelength difference between these two detection methods is significant, so the impurities in the solution are difficult to have a greater impact on both detection methods. Hence, dual-channel signal reading has better robustness, accuracy, and fewer errors than the single-channel signal reading method. 

The color development results of the DPD colorimetric method and the OTO colorimetric method in the microfluidic chip under 490–510 nm and 440–460 nm light irradiation are shown in Figure 6, respectively.

### 2.6. The Construction of LS-BP Algorithm

A LS-BP algorithm was proposed to accurately predict the residual chlorine in water, which can realize dual-channel signal reading. The LS-BP algorithm is based on a least square method and a BP neural network.

The detection of residual chlorine in water through the DPD colorimetric method or the OTO colorimetric method may be affected by impurities in the solution. The impurities probably have a strong absorption capacity for the light of certain wavelengths. This may cause fluctuations in the absorbance of the solution and may lead to inaccurate detection results and deviation of the detection results.

When the DPD colorimetric method and the OTO colorimetric method are used simultaneously to detect residual chlorine, the wavelength difference between these two detection methods is significant, so the impurities in the solution struggle to have a greater impact on both detection methods. Hence, dual-channel signal reading has better robustness, accuracy, and smaller error than the single-channel signal reading method. 

The least squares method is a mathematical optimization method which can find the best function of the data by minimizing the sum of squares of errors and performing prediction. The BP neural network is a multi-layer feedforward neural network, which is mainly characterized by signal forward transmission and error backpropagation [58]. During forward transmission in a neural network, the input signal undergoes layer-by-layer processing from the input layer to the hidden layer. This iterative process continues until the neuron states of the output layer exclusively influence the state of the subsequent layer of neurons. During the process of forward propagation, the input signal is processed layer by layer from the input layer to the hidden layer, until the neuron states of each layer of the output layer only influence the states of the next layer of neurons. When the output from the output layer does not match the expected result, the error signal between them is backpropagated, and the weights are iteratively updated until the error signal reaches the specified minimum.

The LS-BP algorithm structure is shown in Figure 7. The least squares method was used for fitting and the prediction residuals were obtained. The BP neural network was used to train the prediction residuals of the function fitted by the least squares method, and the compensation function of the least squares prediction function was obtained. After adding the compensation function to the original equation, the function with smaller prediction residuals and higher prediction precision was obtained. 

The hidden layer activation function of the BP neural network is a tangent hyperbolic tanh function,
(10)$$f_{1}\left(x\right)=\frac{2}{e^{-2x}+1}-1$$

The activation function of the neural network output layer is the ReLU function,
(11)$$f_{2}\left(x\right)=max(0,x)$$

The BP neural network is a single hidden layer, and the mathematical expression is
(12)$$u_{i}=f_{1}(\sum_{i=1}^{n}v_{ij}x_{i}+\theta_{j}^{u})$$
(13)$$y=f_{2}(\sum_{j=1}^{m}w_{j}u_{i}+\theta^{y})$$

Among them, $x_{i}$ is the input, $u_{i}$ is the hidden layer output, y is the output, $v_{ij}$ is the weight of the *i*-th input variable and the *j*-th neuron, $w_{j}$ is the weight of the *j*-th neuron and the output variable, $\theta_{j}^{u}$ is the threshold of the *j*-th neuron in the hidden layer, $\theta^{y}$ is the threshold of y, $y_{i}$ is the prediction result of the least squares function variance, $y_{i}^{t}$ is the actual result, and $r_{i}$ represents the prediction residuals of the least squares method. LS is the least squares method. Overall, 70% of the samples were used for neural network model training, 15% of the samples were used for neural network model verification, 15% of the samples were used for neural network model detection, the number of input layer nodes was one, the number of hidden layer nodes was selected according to the training effect of different training sets, the number of output layer nodes was one, the input value was the detection voltage, and the output value was the prediction residual of the function fitted by the least squares method. Through the training of the BP neural network, the compensation function was obtained. After processing the function fitted by the least squares method by the compensation function, the final function of detection voltages and residual chlorine concentrations was obtained. 

Mean absolute percentage error (MAPE) is an evaluation index that is commonly used to measure the average percentage error between the predicted value and the true value. It measures the relative error between the predicted value and the true value. 

The average absolute percentage error calculation formula is as follows:(14)$$MAPE=\frac{100\%}{n}\sum_{i=1}^{n}\left|\frac{{\hat{y}}_{i}-y_{i}}{y_{i}}\right|$$
where $y_{i}$ is the true value and ${\hat{y}}_{i}$ is the predicted value.

### 2.7. Characterizations

The constant voltage power supply (Shenzhen Zhongshun Xinneng Battery Co., Ltd., Shenzhen, China) was used to supply power to the portable residual chlorine detection device. The integrated circuit (Zhengzhou Hesheng Electronic Technology Co., Ltd., Zhengzhou, China) was designed to control the portable residual chlorine detection device. The micro peristaltic pumps (Shenzhen Jichuangxing Technology Co., Ltd., Shenzhen, China) were used to control the liquid in the microfluidic chip. The LED constant light source (Shenzhen Fangpu Optoelectronics Co., Ltd., Shenzhen, China) was used to provide light. Filters (Shenzhen Infrared Laser Technology Co., Ltd., Shenzhen, China) were used to obtain light of a specific wavelength. The OLED screen (Zhengzhou Zhongjingyuan Electronic Technology Co., Ltd., Zhengzhou, China) was used to present the detection result. The HC-05 Bluetooth module (Shenzhen Feiyitong Technology Co., Ltd., Shenzhen, China) was used to transfer the test results to the phone. The portable residual chlorine detection device shell and microfluidic chips were prepared by a HALOT-MAGE photocuring printer (Creality 3D Technology Co., Ltd., Shenzhen, China).

The detection liquid for color reaction was configured by the DPD detection reagent and the OTO detection reagent (Hangzhou Luheng Water Quality Detection Co., Ltd., Hangzhou, Zhejiang, China). The three-dimensional structure of the microfluidic chip was designed using SOLIDWORKS 2020 (Dassault Systèmes SOLIDWORKS Corp, Massachusetts, USA) software. The mixing process was simulated by using the finite element analysis software COMSOL Multiphysics 6.1 (COMSOL Inc., Stockholm, Sweden) and its parametric scanning function.

## 3. Results and Discussion

### 3.1. Influence of Microfluidic Chip Structure Parameters on Liquid Mixing Efficiency of Microfluidic Chip

Through the fluid dynamics simulation results of the liquid mixing in microfluidic chip under different parameters, the influence of the channel amplitude *A*, the channel width *α*, and the angular frequency *ω* on liquid mixing efficiency was obtained.

The influence of channel amplitude *A* on liquid mixing efficiency is shown in Figure 8. When the channel width *α* is 1 mm and the channel angular frequency *ω* is 1 rad·s^−1^, the liquid mixing efficiency increases accordingly as the channel amplitude *A* increases from 0.5 mm to 2 mm. 

The influence of the channel width *α* on liquid mixing efficiency is shown in Figure 9. When channel amplitude *A* is 1 mm and channel angular frequency *ω* is 1 rad·s^−1^, the liquid mixing efficiency decreases accordingly as channel width *α* increases from 0.5 mm to 2 mm.

The influence of the channel angular frequency *ω* on liquid mixing efficiency is shown in Figure 10. When channel amplitude *A* is 1 mm and channel width *α* is 1 mm, the liquid mixing efficiency increases accordingly as the channel angular frequency *ω* increases from 0.5 rad·s^−1^ to 2 rad·s^−1^.

In the case of analyzing three factors separately, when channel amplitude *A* is 2 mm, channel width *α* is 0.5 mm, and channel angular frequency *ω* is 2 rad·s^−1^, the liquid mixing efficiency is relatively high, which is beneficial for liquid mixing in microfluidic chips. When the channel amplitude and channel angular frequency increase, the degree of channel bending increases separately, and the Reynolds number increases. Consequently, as the Reynolds number increases, the mixing efficiency also improves. When the channel width increases, the channel widens, the liquid flow rate decreases, the Reynolds number decreases, and the mixing efficiency decreases. This result indicates that within a certain range, an increase in channel amplitude and channel angular frequency has a positive effect on liquid mixing in microfluidic chips, while an increase in channel width has a negative effect.

Through the fluid dynamics simulations, the comprehensive influence of channel amplitude *A*, channel width *α*, and channel angular frequency *ω* on liquid mixing efficiency was obtained. The influence of these factors on mixing efficiency is shown in Figure 11, which represents the mixing index at different values of each factor at the 5 mm cross-section, 10 mm cross-section, 15 mm cross-section, and outlet cross-section. It generally indicates that under the combined influence of three factors on the mixing efficiency of microfluidic chips, larger channel amplitude *A*, smaller channel width *α,* and channel angular frequency *ω* lead to higher mixing efficiency. After specific analysis and calculation by software, a set of parameters was obtained for the microfluidic chip with the highest liquid mixing efficiency, with channel amplitude *A* of 1.8 mm, channel width *α* of 0.7 mm, and channel angular frequency *ω* of 0.7 rad·s^−1^.

The fluid dynamics simulation result of the liquid mixing in the mixing channel was obtained, as shown in Figure 12. The liquid mixing was mainly completed before 5 mm. The result indicates that the microfluidic chip has high liquid mixing efficiency.

The mixing index of each cross-section in microfluidic chip with the highest liquid mixing efficiency is shown in Figure 13. From 0 mm to 10 mm, the mixing index shows a rapid decreasing trend, which has a good effect on liquid mixing. From 10 mm to 45 mm, the mixing index remains approximately unchanged, and it can be considered that the liquid mixing is completed. This result indicates that this microfluidic chip has high mixing efficiency and can achieve efficient and rapid mixing of two-phase flow.

The microfluidic chip shape with the highest liquid mixing efficiency was fabricated by a 3D printing method, as shown in Figure 14. The length, width, and height of the prepared microfluidic chip is 77 mm, 20 mm and 4 mm, respectively. It has good machining uniformity. The fabricated microfluidic chip has the highest liquid mixing efficiency, which provides a basis for rapid and convenient detection of residual chlorine in water.

The design, parameter optimization, and preparation of microfluidic chips can achieve efficient mixing of two-phase flow and provide a foundation for rapid, low-cost, and convenient detection of the detection device. This microfluidic chip can be used for two-phase flow mixing in other devices, effectively reducing the volume and cost of the devices and achieving rapid and efficient mixing.

### 3.2. Calibration and Evaluation of the LS-BP Algorithm

The DPD colorimetric method and the OTO colorimetric method were used, respectively, to predict residual chlorine in water. The average absolute percentage errors of the prediction functions fitted by the least squares method were 2.294% and 1.975%. The BP neural network was used to process the residual predictions. The average absolute percentage errors of the prediction functions were 2.104% and 1.738%.

The DPD colorimetric method and the OTO colorimetric method were used simultaneously to predict residual chlorine in water. The detection voltages of these two methods were weighted to obtain the weighted voltages. The least squares method was used to fit the weighted voltages and the concentrations of the liquid to be detected. The actual results and the curve fitted results of the detected residual chlorine concentration by the least squares method are shown in Figure 15. The actual results were drawn every four intervals in the figure, and the total number of actual results was 100. The average absolute percentage error (MEAP) of the prediction function is 1.637%, and the standard deviation (SD) is less than 0.21. The horizontal axis voltage value is the average voltage obtained from three detections of residual chlorine solutions with different concentrations. The error bar is mainly caused by the prediction of residual chlorine concentration using three voltages obtained from detection. There was a certain deviation between the actual results and the fitted results.

The BP neural network was used to process the prediction residuals of the function fitted by the least squares method, and the compensation function can be obtained. After processing was performed via the BP neural network, a more accurate prediction function was obtained. The actual results and the curve fitted results of the detected residual chlorine concentration by the combination of the least squares method and the BP neural network are shown in Figure 16. The actual results were drawn every four intervals in the figure, and the total number of the actual results was 100. The average absolute percentage error (MEAP) of the prediction function is 0.24%, and the standard deviation (SD) is less than 0.21. The deviation between the fitted results and the actual results was very small and the accuracy was high.

As shown in Figure 17, the prediction residuals of the function fitted by the least squares method before and after the BP neural network processing were compared. AVG in the figure presents the average value, and MS in the figure presents the mean squared error. The average residuals of the functions were 1.8 × 10^−3^ mg·L^−1^ and 1.019 × 10^−4^ mg·L^−1^. The mean squared errors of the residuals of the functions were 2.1 × 10^−3^ and 1.781 × 10^−4^. The prediction residuals obtained after the BP neural network processing were significantly reduced.

The average prediction residual obtained by the DPD colorimetric method or the OTO colorimetric method is larger than the average prediction residual obtained by the combination of these two colorimetric methods. The error of the prediction function fitted by the least squares method is larger than the error of the prediction function fitted by the least squares method and the BP neural network. Therefore, the LS-BP algorithm can improve the accuracy of the prediction results and reduce prediction errors.

The function fitted by the least squares method for the residual chlorine prediction is
(15)y=57.25x−17.2
(16)$$x=0.5x_{1}+0.5x_{2}$$
where x is the weighted voltage value, $x_{1}$ is the voltage value detected by the DPD method, $x_{2}$ is the voltage value detected by the OTO method, and *y* is the predicted concentration.

The compensation function obtained by the BP neural network is
(17)$$y_{BP}\left(x^{\prime }\right)=f_{2}\left[\sum_{j=1}^{101}{\omega_{j}f}_{1}\left(v_{j}x^{\prime }+\theta_{j}^{u}\right)+\theta^{y}\right]$$
where the values of $\theta^{y}$, $v_{j}$, $\theta_{j}^{u}$, $\omega_{j}$ can be found in Appendix A.

The prediction function after the compensation function processing is
(18)$$y=57.25x-y_{BP}^{\prime }\left(x^{\prime }\right)-17.2$$
where $x^{\prime }$ is the normalized result of x, which can be calculated as follows:(19)$$x^{\prime }=\left(x-0.3\right)\times 11.4286$$
and $y_{BP}^{\prime }\left(x^{\prime }\right)$ is the result of reverse normalization of $y_{BP}\left(x^{\prime }\right)$, which can be calculated as follows:(20)$$y_{BP}^{\prime }\left(x^{\prime }\right)=\frac{y_{BP}\left(x^{\prime }\right)+1}{9.9931}-0.0968$$

The proposal of the LS-BP algorithm provides necessary conditions for the detection device to achieve accurate and rapid detection. This algorithm also has good accuracy and reliability for detecting other harmful substances in water, and further related research will be conducted in the future.

### 3.3. Performance Evaluation the Portable Residual Chlorine Detection Device 

The residual chlorine solutions of different concentrations were measured three times, and the detection results of the blank samples are shown in Table 1. The blank samples were measured 10 times, and the detection results of different concentrations of residual chlorine standard solution are shown in Table 2.

Based on the detection results, the limit of detection and the relative standard deviation of the portable residual chlorine detection device were calculated. The portable residual chlorine detection device is 110 mm long, 110 mm wide, and 60 mm high, respectively, which can meet the need for portability. Sargazi et al. [59], Uriarte et al. [60], and Dou et al. [24] studied rapid residual chlorine detection devices based on smartphones. Yen et al. studied a portable paper-based chemical sensor for residual chlorine detection [23]. Kato et al. studied an all-solid residual chlorine sensor for monitoring tap water quality [61]. Huangfu et al. studied a μPAD for simultaneous monitoring of free chlorine [62]. Xiong et al. [63] and Lu et al. [64] studied fluorescence detection for residual chlorine detection. The portable residual chlorine detection device in this study was compared with the detection devices and methods mentioned above, and the results of the comparison are shown in Table 3. 

The comparison results show that compared to other devices and detection methods, the portable residual chlorine detection device has relatively high detection accuracy and reliability, fast detection speed, and low consumption of detection reagents. The construction and maintenance costs of the detection device are low, and maintenance mainly involves the replacement of microfluidic chips and the consumption of detection reagents. Overall, the portable residual chlorine detection device is low-cost. 

Therefore, the portable residual chlorine detection device can achieve accurate, rapid, low-cost and convenient detection of residual chlorine in water. It can meet the needs of residual chlorine detection in related fields, such as drinking water, swimming pools, and aquaculture. The portable residual chlorine detection device can achieve accurate, rapid, low-cost and convenient detection of the residual chlorine in water, filling the gap in the residual chlorine detection field, but there are still some unresolved issues. There are some difficulties when it comes to mass production of the portable residual chlorine detection device. Each sensor used may have certain deviations and needs to be calibrated before use, otherwise it will cause significant deviations in the detection results. Calibration will take a lot of time and manpower. There is still room for optimization in the volume and quality of the detection device. The surface quality of microfluidic chips prepared using a photopolymerization 3D printer cannot be guaranteed; therefore, further exploration of preparation factors is needed. Therefore, in future research, efforts will be made to address these issues.

## 4. Conclusions

A portable residual chlorine detection device with a combination of microfluidic chips and deep learning algorithms was developed. The portable residual chlorine detection device can achieve accurate, rapid, low-cost, and convenient detection of the residual chlorine in water, filling the gap in the residual chlorine detection field. Here are some detailed conclusions.

(1)A microfluidic chip that can achieve efficient mixing of two-phase flow was studied. The results indicate that channel amplitude *A*, channel width *α*, the channel angular frequency *ω* have an impact on mixing efficiency. The increase in channel amplitude *A* and channel width *α* is beneficial for improving the mixing efficiency, and the increase in channel width is not beneficial for improving the mixing efficiency. When channel amplitude *A* is 1.8 mm, channel width *α* is 0.7 mm, and channel angular frequency *ω* is 0.7 rad·s^−1^, the microfluidic chip has good mixing efficiency. This microfluidic chip can also be used for liquid mixing in other detection devices, reducing device volume and cost and achieving efficient and fast mixing.(2)An LS-BP algorithm was proposed, which is based on the least squares method and the BP neural network. The LS-BP algorithm was used to predict the residual chlorine concentration in water, and it has good accuracy. The average absolute percentage error of the prediction result is 0.24%, the average of the prediction residuals is 1.781 × 10^−4^ mg·L^−1^, and the variance of the prediction residuals is 1.019 × 10^−4^. This algorithm is also applicable to the detection of other substances in water and still has good detection accuracy and reliability, which will be further confirmed in future research.(3)The limit of detection of the portable residual chlorine detection device is 0.01 mg·L^−1^, the relative standard deviation is 3.2%, the detection reagent is 50 s, the detection liquid consumption is 5 mL, and the construction and maintenance costs are low. Compared with other residual chlorine detection devices and methods, the portable residual chlorine detection device has relatively high detection accuracy, fast detection speed, a low cost, and is more convenient. The portable residual chlorine detection fills the gap in the absence of a device that can accurately, rapidly, and conveniently detect residual chlorine in water at low cost. It can also be used to detect residual chlorine in other types of water, such as drinking water, swimming pools, and aquaculture. It can also be used to detect residual chlorine in water, such as drinking water, swimming pools, and aquaculture.

## Figures and Tables

**Figure 1 micromachines-15-01045-f001:**
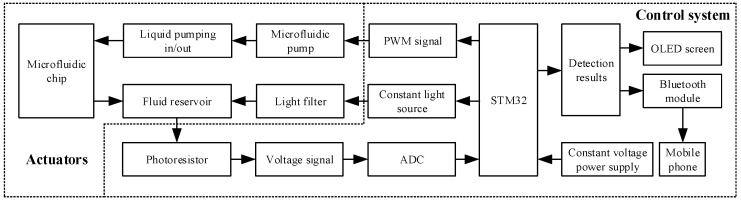
The working principle of the portable residual chlorine detection device.

**Figure 2 micromachines-15-01045-f002:**
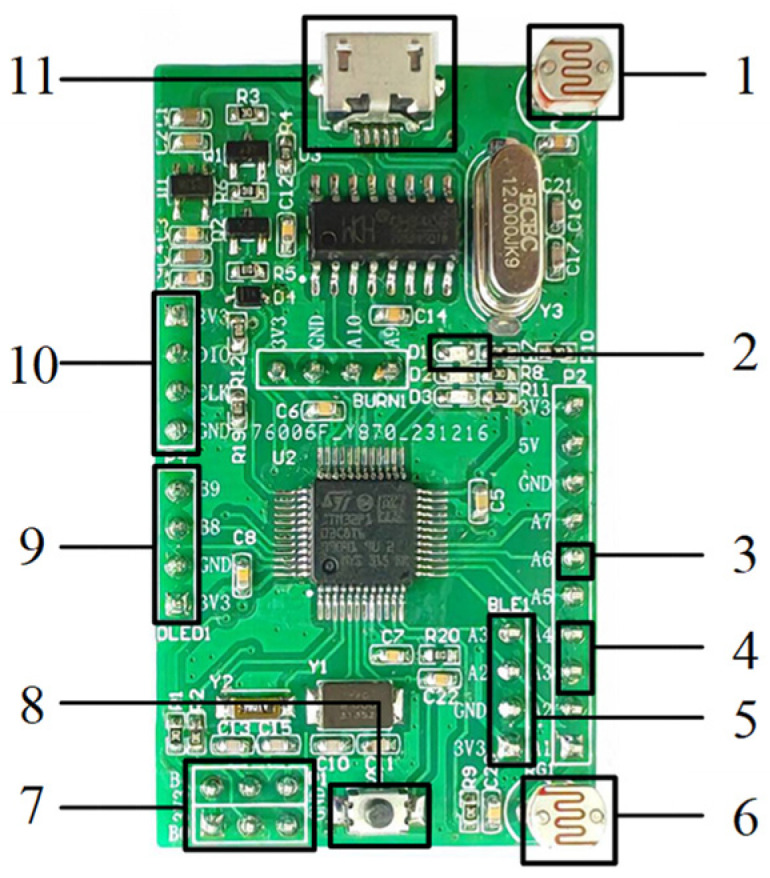
The designed integrated circuit. 1. Photoresistor; 2. power indicator light; 3. constant light source control pin; 4. micro peristaltic pump control pin; 5. Bluetooth module control pin; 6. photoresistor; 7. BOOT pin; 8. reset button; 9. OLED screen control pin; 10. download port; 11. USB power port.

**Figure 3 micromachines-15-01045-f003:**
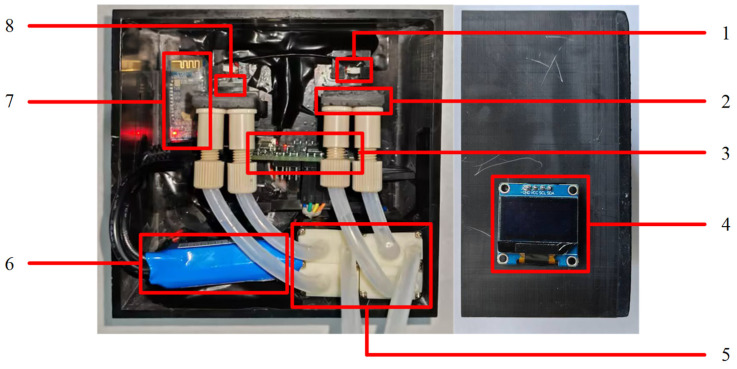
Composition of the portable residual chlorine detection device. 1. LED constant light source; 2. microfluidic chip; 3. integrated circuitry; 4. OLED screen; 5. micro peristaltic pumps; 6. constant voltage power supply; 7. Bluetooth module; 8. light filter.

**Figure 4 micromachines-15-01045-f004:**
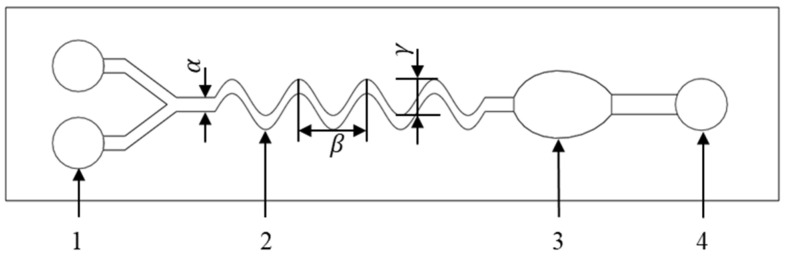
Parameter design of mixing channel structure for microfluidic chips. 1. Inlet; 2. mixing channel; 3. observation port; 4. outlet.

**Figure 5 micromachines-15-01045-f005:**
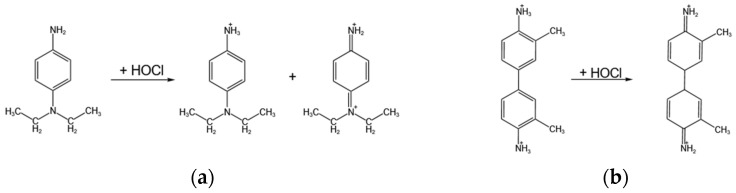
The principles of these two colorimetric methods. (**a**) The chemical reaction equation of the DPD colorimetric method principle. (**b**) The chemical reaction equation of the OTO colorimetric method principle.

**Figure 6 micromachines-15-01045-f006:**
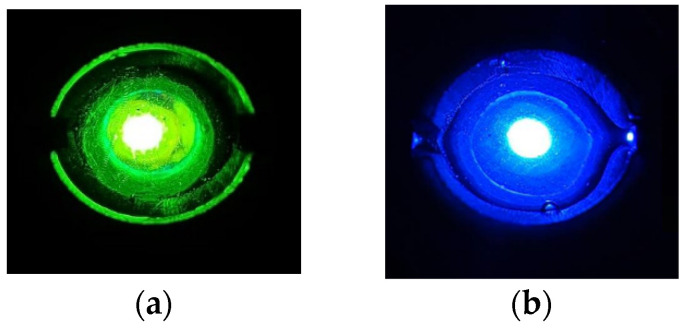
Color development of these two detection methods in microfluidic chip. (**a**) Color development of the DPD colorimetric method in microfluidic chip under 490–510 nm light irradiation. (**b**) Color development of the OTO colorimetric method in microfluidic chip under 440–460 nm light irradiation.

**Figure 7 micromachines-15-01045-f007:**
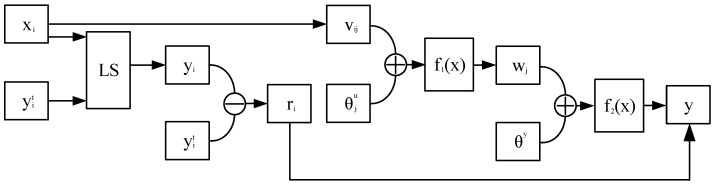
The LS-BP algorithm structure.

**Figure 8 micromachines-15-01045-f008:**
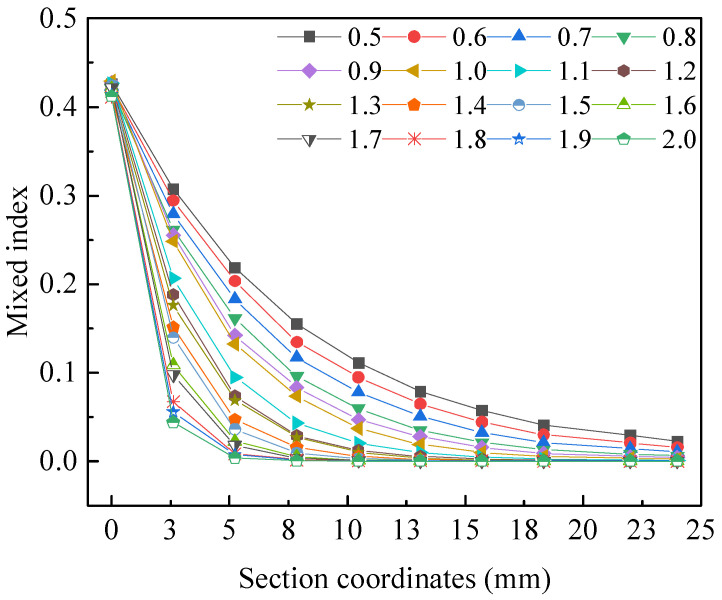
The influence of channel amplitude *A* on liquid mixing efficiency.

**Figure 9 micromachines-15-01045-f009:**
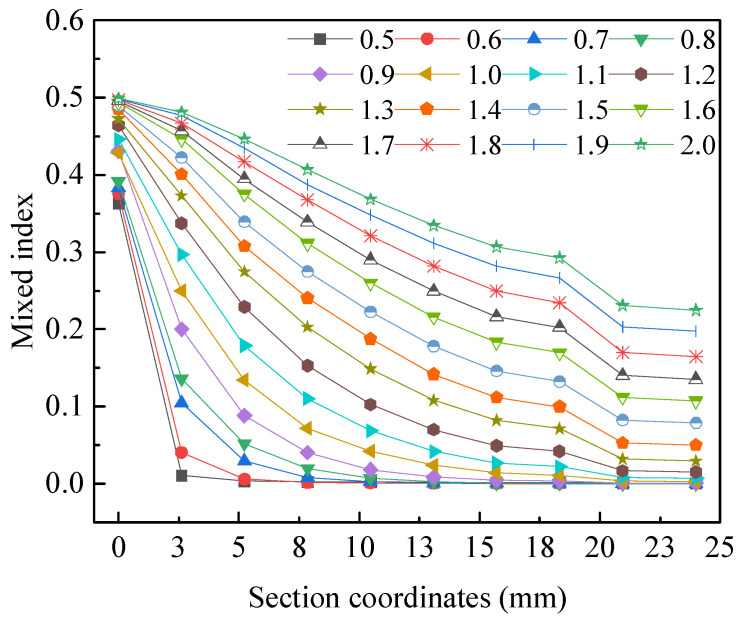
The influence of the channel width *α* on liquid mixing efficiency.

**Figure 10 micromachines-15-01045-f010:**
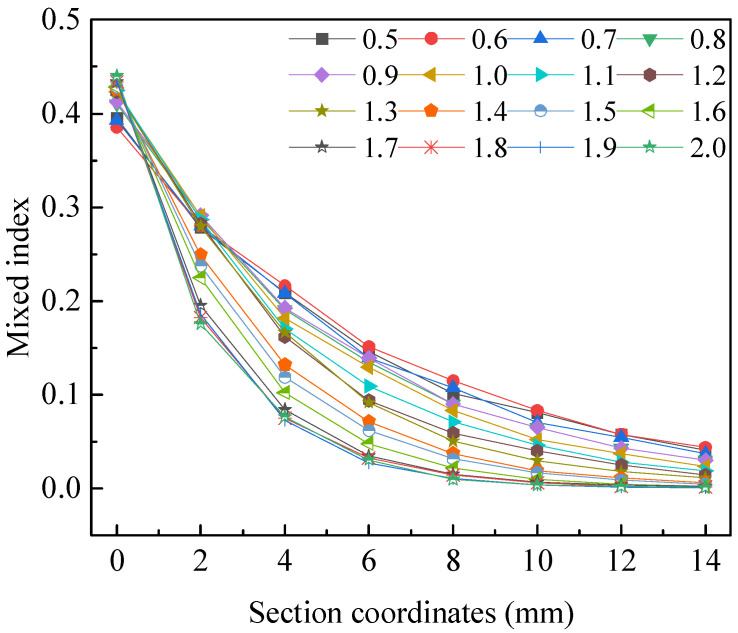
The influence of the channel angular frequency *ω* on liquid mixing efficiency.

**Figure 11 micromachines-15-01045-f011:**
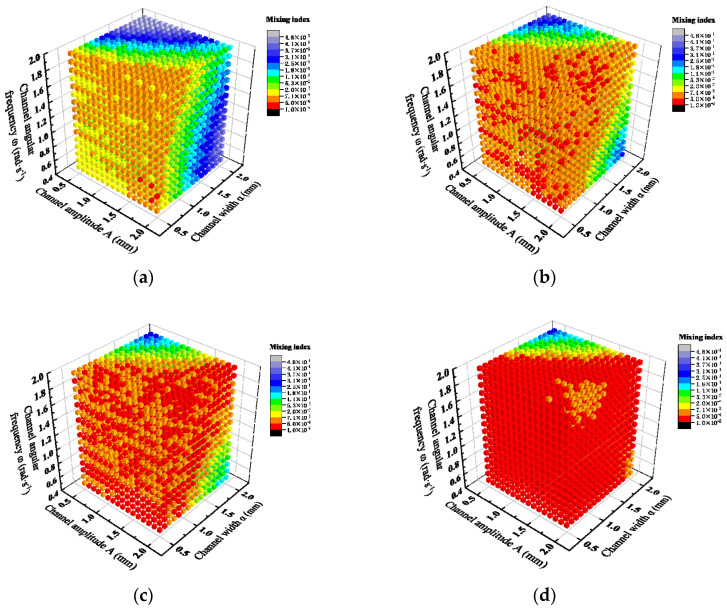
The influence of three factors on liquid mixing efficiency. (**a**) Mixing efficiency at 5 mm cross-section under different values of each factor. (**b**) Mixing efficiency at 10 mm cross-section under different values of each factor. (**c**) Mixing efficiency at 15 mm cross-section under different values of each factor. (**d**) Mixing efficiency at outlet cross-section under different values of each factor.

**Figure 12 micromachines-15-01045-f012:**
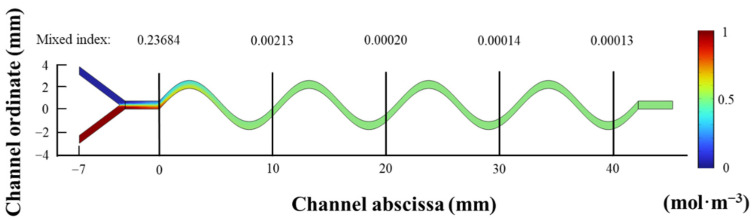
The fluid dynamics simulation result of the liquid mixing in the mixing channel.

**Figure 13 micromachines-15-01045-f013:**
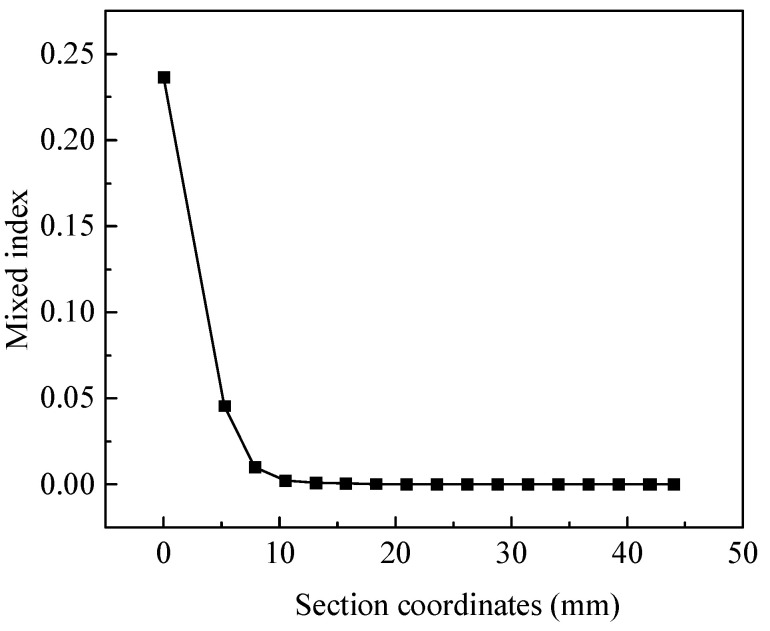
The cross-section mixing index of microfluidic chip with the highest liquid mixing efficiency.

**Figure 14 micromachines-15-01045-f014:**
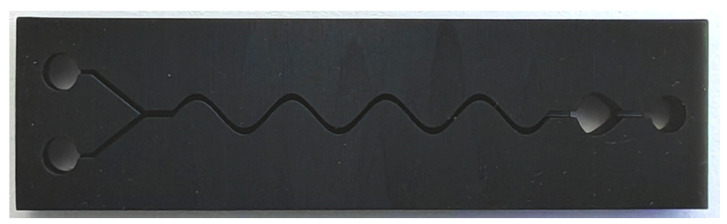
The microfluidic chip with the highest liquid mixing efficiency fabricated by 3D printing method.

**Figure 15 micromachines-15-01045-f015:**
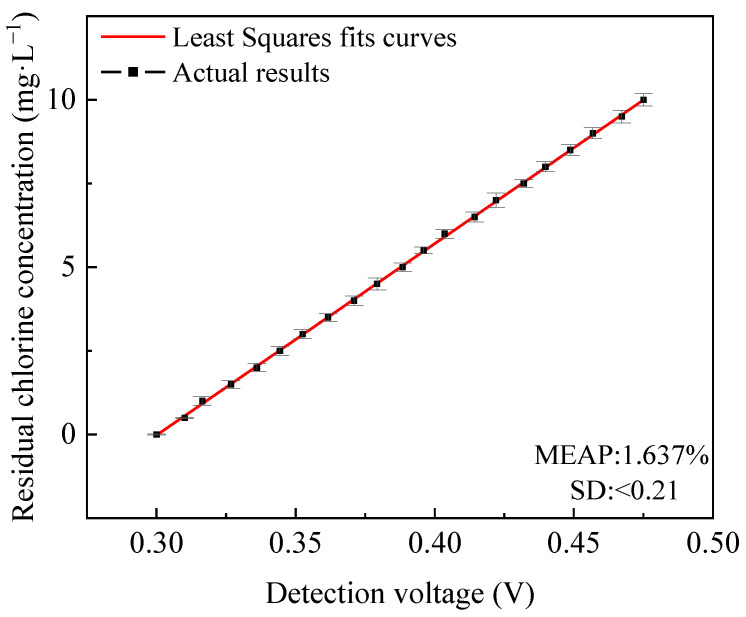
The actual results and the curve fitted results of the detected residual chlorine concentration by the least squares method.

**Figure 16 micromachines-15-01045-f016:**
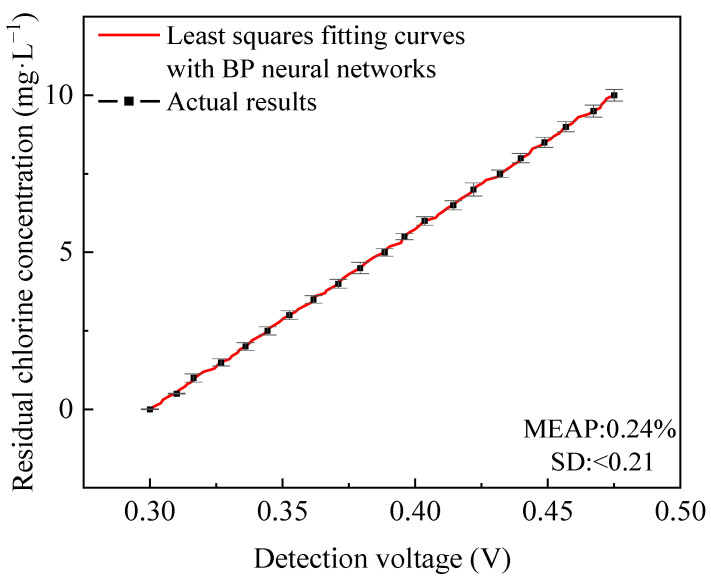
The actual results and the curve fitted results of the detected residual chlorine concentration by the combination of the least squares method and the BP neural network.

**Figure 17 micromachines-15-01045-f017:**
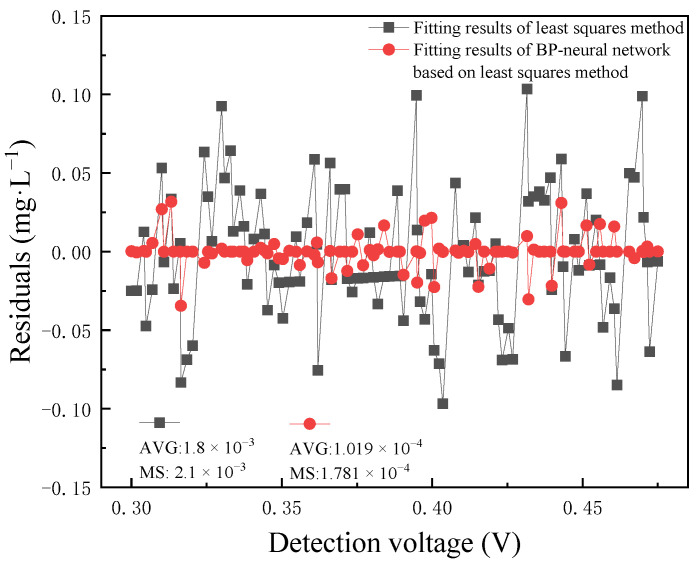
The comparison between the prediction residuals of the prediction function fitted by the least squares method before and after processed by the BP neural network.

**Table 1 micromachines-15-01045-t001:** Detection results of blank samples.

Sample Number	Detection Results of Blank Samples (mg·L^−1^)
1	0.012
2	0.011
3	0.008
4	0.007
5	0.009
6	0.008
7	0.011
8	0.014
9	0.012
10	0.011

LOD is 0.01 mg·L^−1^.

**Table 2 micromachines-15-01045-t002:** Detection results of different concentrations of residual chlorine standard solution.

Concentration of Residual Chlorine Standard Solution (mg·L^−1^)	First Detection	Second Detection	Third Detection
1	1.075	1.067	1.013
2	2.116	2.054	1.987
3	3.097	3.168	3.029
4	4.133	4.197	4.031
5	5.141	5.236	4.938
6	5.841	6.035	6.138
7	7.275	7.232	6.931
8	8.027	8.119	7.891
9	9.217	9.128	8.831
10	9.832	10.241	10.145

RSD is less than 3.2%.

**Table 3 micromachines-15-01045-t003:** Comparison between the self-developed portable residual chlorine detection device and other residual chlorine detection devices and methods.

Item	Limit of Detection (mg·L^−1^)	Relative Standard Deviation	Detection Range (mg·L^−1^)	Detection Time (min)	Consumption of Detection Reagents (mL)
Sargazi et al., + 2020 [59]	0.05	8.75%	1–4	2	5
Uriarte et al., + 2021 [60]	0.006	4.6%	0.02–0.5	--	5
Dou et al., + 2020 [23]	0.161	--	0.56–9.8	30	5
Yen et al., + 2019 [24]	0.18	--	0.1–500	5	--
Kato et al., + 2017 [61]	0.1	--	0.3–1	4	--
Huangfu et al., + 2019 [62]	0.2	--	0.2–5	--	10
Xiong et al., + 2015 [63]	0.035	4.2	0.056–56	20	0.12
Lu et al., + 2016 [64]	0.028	--	0.035–10.5	5	100
The portable residual chlorine detection device in this study	0.01	3.2%	0–10	0.8	5

## Data Availability

The data that support the findings of this study are available from the corresponding author upon reasonable request.

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
