# Peer review of "Development of a Portable Residual Chlorine Detection Device with a Combination of Microfluidic Chips and LS-BP Algorithm to Achieve Accurate Detection of Residual Chlorine in Water"

_micromachines, 2024, doi:10.3390/mi15081045_

Round 1

Reviewer 1 Report

Comments and Suggestions for Authors

The authors presents  a novel portable device for detecting residual chlorine in drinking water, leveraging microfluidic technology to enhance measurement accuracy and efficiency. The device integrates advanced algorithms to process data, ensuring rapid results that are crucial for public health monitoring. The microfluidic chip utilized in the device facilitates efficient liquid mixing, as demonstrated by fluid dynamics simulations, which optimize the detection process. The research highlights the importance of maintaining appropriate chlorine levels to prevent microbial contamination while avoiding excessive concentrations that can pose health risks. The authors conducted extensive experiments to validate the device's performance, showcasing its reliability in various water samples. Additionally, the study discusses the implications of residual chlorine detection in ensuring safe drinking water, particularly in regions with limited access to water quality monitoring. Overall, this innovative approach represents a significant advancement in water quality assessment technologies. Future work may focus on further miniaturization and integration of the device for field applications.

Several comments can be made on the article:

1. Is the designed integrated circuit shown in Figure 2 the authors' development? If it is a typical device, its manufacturer and name must be indicated

2. In Figure 3, the microfluidic chips are located above the electronic board. How well does this solution protect against leaks?

3. What was the repeatability of the results? How many times was the experiment carried out? What is the measurement error? It is not shown in graphs 13 and 14.

After these  minor changes, the article can be published.

Author Response

Author response: On behalf of my co-authors, we thank you very much for giving us opportunity to revise our manuscript, we appreciate the reviewer very much for your positive and constructive comments and suggestions on our manuscript. Those comments are all valuable and very helpful for revising and improving our paper, as well as the important guiding significance to our researches. We have studied comments carefully and have made correction which we hope meet with approval.

Based on the reviewer's comments, we improved and modified our research.

Response to Reviewer#1:

Comments 1: Is the designed integrated circuit shown in Figure 2 the authors' development? If it is a typical device, its manufacturer and name must be indicated.

Respnse 1: The authors are very grateful to the reviewers for the revision comment. The designed integrated circuit shown in Figure 2 was developed by the authors, and PCB circuit drawing, the selection of electronic components, etc. were all personally designed by the authors. However, due to the inability to provide some experimental equipment, the processing of integrated circuits was completed by relevant enterprises, and the specific enterprises was proposed in Section 2.6. Thank you again sincerely for the feedback.

Comments 2: In Figure 3, the microfluidic chips are located above the electronic board. How well does this solution protect against leaks?

Respnse 2:

The authors greatly appreciate the reviewer's comments and understand your concerns.

Firstly, once liquid leakage occurs, it will cause serious problems, so the authors also pay attention to this design issue.

Secondly, in this design, the shortest channel flow path and the implementation of related functions are given priority, but relevant measures have also been taken to prevent liquid leakage.The upper and lower surfaces of microfluidic chips were encapsulated with nano paste, resulting in a good encapsulation effect. Nano paste has good adhesion, stable chemical properties, and low friction coefficient, with little impact on liquids in microfluidic chips. The microfluidic chips were affixed to the joint base with 502 adhesive, providing a secure and leak-free seal. The connection between the joint base and the joint utilized threading, ensuring reliability and preventing any liquid leakage. In addition, during multiple experiments using the device, no liquid leakage was observed in the microfluidic chips, demonstrating the excellent sealing performance. Thirdly, in the subsequent design, authors will optimize the design layout based on the reviewer's comments and apply waterproof treatment to the circuit. The authors have updated the relevant content in the manuscript to provide a detailed explanation of their methods for preventing leaks and test results.

Action 2: The following content was added to the manuscript.

Section 2.1

The microfluidic chips in the detection device were encapsulated with nano past, the microfluidic chips were affixed to the joint base with 502 adhesive, the connection between the joint base and the joint utilized threading, preventing any liquid leakage. And during multiple experiments using the device, no liquid leakage was observed in the microfluidic chips, demonstrating the excellent sealing performance.

Comments 3: What was the repeatability of the results? How many times was the experiment carried out? What is the measurement error? It is not shown in graphs 13 and 14.

Respnse 3:

The authors sincerely for your questions regarding the repeatability of the results and the experimental details.

  • The repeatability of the experiment was evaluated using standard deviation, which was less than 0.21.
  • The experiment was conducted 3 times to ensure robustness and reliability in our findings.
  • The author added error bars for measurement errors in the figure to indicate the measurement errors present in the experiment.

Action 3:

The average absolute percentage error (MEAP) of the prediction function is 1.637%, and the standard deviation (SD) is less than 0.21. The horizontal axis voltage value is the average voltage obtained from three detections of residual chlorine solutions with different concentrations. The error bar is mainly caused by the prediction of residual chlorine concentration using three voltages obtained from detection.

Figure 15. The actual results and the curve fitted results of the detected residual chlorine concentration by the least squares method.

The average absolute percentage error (MEAP) of the prediction function is 0.24%, and the standard deviation (SD) is less than 0.21.

Figure 16. The actual results and the curve fitted results of the detected residual chlorine concentration by the combination of the least squares method and the BP-neural network.

Reviewer 2 Report

Comments and Suggestions for Authors

In this manuscript, a portable residual chlorine detection device with a combination of microfluidic chips and deep learning algorithms was developed. The portable residual chlorine detection device can achieve accurate, rapid, low-cost and convenient detection of the residual chlorine in water, filling the gap in residual chlorine detection field. The paper contains interesting results. However, the presentation of the study can be improved. All the sections are way longer than they can be, which is a general comment for the manuscript. Some contents are highly repetitive, which could be more concise and logical to make it more readable. There are few deficiencies which the authors should address before publishing.

1. Introduction, Lines 57-113: Avoid simply listing of literatures, and it would be better to show the advantages of microfluidic chips, deep learning algorithms and dual-channel signal reading separately (i.e. separate paragraphs).

2. Experiments, Figure 3:What is the size and mass of the device, this is nice to know given the authors are suggesting portability.

3. Experiments, Tables 1-4: Tables contains only a single row of information are not recommended.

4. Experiments, Lines 277-299: The subtitle is The construction of LS-BP Algorithm, but there are 3 paragraphs about two colorimetric methods in the middle. Besides, I suggest to introduce the basic information of LS-BP algorithm in the Introduction section instead of the Experiments section.

5. Result and discussion, Lines 372-402: The descriptions of the channel amplitude, channel width, and channel angular frequency are highly repetitive, which could be further integrated. Besides, the author mentioned that when investigating one factor, the other two factors remain constant. Whats the exact value? How they influence each other? Why the best channel amplitude A is 2 mm, the channel width α is 0.5 mm, and the channel angular frequency ω is 2 rad s-1 in the single factor analysis, while the comprehensive analysis obtained inconsistent results (1.8 mm, 0.7mm, and 0.7 rad s-1, respectively)?

6. Conclusion: The limitations of the present study should be addressed.

Comments on the Quality of English Language

Some contents are highly repetitive, which could be more concise and logical to make it more readable. 

Author Response

Author response: On behalf of my co-authors, we thank you very much for giving us opportunity to revise our manuscript, we appreciate the reviewer very much for your positive and constructive comments and suggestions on our manuscript. Those comments are all valuable and very helpful for revising and improving our paper, as well as the important guiding significance to our researches. We have studied comments carefully and have made correction which we hope meet with approval.

Based on the reviewer's comments, we improved and modified our research.

Response to Reviewer#2:

Comments 1: Introduction, Lines 57-113: Avoid simply listing of literatures, and it would be better to show the advantages of microfluidic chips, deep learning algorithms and dual-channel signal reading separately (i.e. separate paragraphs).

Respnse 1: The authors greatly appreciate your constructive suggestion to avoid simply listing literature and instead highlight the advantages of microfluidic chips, deep learning algorithms, and dual-channel signal reading separately. In the revised manuscript, the authors will restructure this section and add relevant references to present distinct paragraphs that explicitly discuss the advantages of each component, thereby improving the clarity and focus of our introduction.

Action 1:

Section 1

In recent years, microfluidic systems have attracted widespread research attention in fields such as analytical chemistry and rapid detection, as they provide a miniaturized platform for traditional analytical techniques. Compared to traditional methods, microfluidic systems allow for faster and lower cost analysis using fewer samples and reagents [30, 31]. Microfluidic chips can be used to build portable detection devices for rapid detection of harmful substances in water bodies. The rapid detection device undoubtedly attracts the favor of users. Its operation is simple and can be used on-site and simultaneously, while microfluidic chips can meet the needs of rapid detection [32].

Based on the above analysis, microfluidic chips have shown great potential in the detection of harmful substances in solutions.

At present, detection devices usually adopt single-channel signal reading method. Although this method is easy to operate, its detection accuracy is susceptible to external interference. The dual-channel signal reading method can effectively reduce the interference generated by the outside world and has better detection accuracy [38].

The above research indicates that dual-channel signal reading method can effectively reduce the impact of external interference on detection results, and can also improve detection accuracy and precision.

Deep learning algorithms have received widespread attention in recent years and have been widely applied in various fields. They have also shown great potential in the field of food inspection [44].

References

[30] Simon Song; Kuen Yong Lee. Polymers for Microfluidic Chips. Macromolecular Research. 2006, 14, 121-128.

[31] Myrto Kyriaki Filippidou; Aris Ioannis Kanaris; Evangelos Aslanidis; Annita Rapesi; Dimitra Tsounidi; Sotirios Ntouskas; Evangelos Skotadis; George Tsekenis; Dimitris Tsoukalas; Angeliki Tserepi; andStavros Chatzandroulis. Integrated Plastic Microfluidic Device for Heavy Metal Ion Detection. Micromachines. 2023, 14(8), 1595.

[32] Minglu Wang; Ying Wang; Xiangyang Li; Hongyan Zhang. Development of a photothermal-sensing microfluidic paper-based analytical chip (PT-Chip) for sensitive quantification of diethylstilbestrol. Food Chemistry. 2023, 402, 134128.

[38] Zhiwei Lu; Jun Qin; Chun Wu; Jiajian Yin; Mengmeng Sun; Gehong Su; Xianxing Wang; Yanying Wang; Jianshan Ye; Tao Liu; Hanbing Rao; Lin Feng. Dual-channel MIRECL portable devices with impedance effect coupled smartphone and machine learning system for tyramine identification and quantification. Food Chemistry. 2023, 429, 136920.

[44] Weiwen He; Hongyuan He; Fanglin Wang; Shuyue Wang; Rulin Lyu. Non-destructive detection and recognition of pesticide residues on garlic chive (Allium tuberosum) leaves based on short wave infrared hyperspectral imaging and one-dimensional convolutional neural network. Journal of Food Measurement and Characterization. 2021, 15, 4497-4507.

Comments 2: Experiments, Figure 3: What is the size and mass of the device, this is nice to know given the authors are suggesting portability.

Respnse 2: The authors thank you for your comment regarding the size and mass of the device used in the experiments, particularly in relation to its suggested portability. The device measures 110mm in length, 110mm in width, and 60mm in height, and its mass is approximately 900 grams. The authors agree that providing this information is crucial for understanding the practical implications of the findings, especially concerning portability. The authors will ensure to include these details in the revised manuscript to enhance clarity and completeness.

Action 2: The following content was added to the manuscript.

Section 2.1

The length, width and height of the portable residual chlorine detection device is 110 mm long, 110 mm wide and 60 mm, respectively high, which can meet the need of portability.

Comments 3: Experiments, Tables 1-4: Tables contains only a single row of information are not recommended.

Respnse 3: The comments of the reviewers are very helpful to the authors, and the authors thank you for the feedback. The authors acknowledge the recommendation regarding Tables 1-4. In response, the authors will revise the tables to include additional rows of information, providing more comprehensive data presentation.

Action 2: The following content was added to the manuscript.

Section 2.2

The portable residual chlorine detection device was used to detect residual chlorine standard solutions ranging from 1 mg·L-1 to 10 mg·L-1, with three measurements taken every 1 mg·L-1, and the device was used to detect the blank samples 10 times to obtain relevant data and calculate LOD and RSD.

Section 2.4

The parameter optimization intervals are: channel amplitude A from 0.5 mm to 2 mm, channel width α from 0.5 mm to 2 mm, and channel angular frequency ω from 0.5 rad·s-1 to 2 rad·s-1.

Section 3.1

A set of parameters was obtained for the microfluidic chip with the highest liquid mixing efficiency, with channel amplitude A of 1.8 mm, channel width α of 0.7 mm, and channel angular frequency ω of 0.7 rad·s-1.

Section 3.3

Table 1. Detection results of blank samples

Sample number

Detection results of blank samples (mg·L-1)

1

0.012

2

0.011

3

0.008

4

0.007

5

0.009

6

0.008

7

0.011

8

0.014

9

0.012

10

0.011

LOD is 0.01 mg·L-1

Comments 4: Experiments, Lines 277-299: The subtitle is “The construction of LS-BP Algorithm”, but there are 3 paragraphs about “two colorimetric methods” in the middle. Besides, I suggest to introduce the basic information of LS-BP algorithm in the Introduction section instead of the Experiments section.

Respnse 4: The authors thank you for your valuable feedback regarding the organization of the manuscript. Regarding the subtitle “The construction of LS-BP Algorithm” and the paragraphs about “two colorimetric methods” in the Experiments section (Lines 277-299), they acknowledge the need for clarity and coherence. In the revised manuscript, the authors will ensure that the content under this subtitle focuses exclusively on the LS-BP Algorithm, providing a cohesive description without digression into unrelated methods. Furthermore, the authors appreciate your suggestion to introduce basic information about the LS-BP algorithm in the Introduction section rather than in the Experiments section. They agree that this adjustment will enhance the logical flow of the manuscript. In the revised version, they will relocate the essential details about LS-BP to the Introduction section to provide readers with a foundational understanding early on. These changes will improve the overall structure and coherence of the manuscript, aligning with your insightful recommendations. Thank you once again for your constructive comments.

Action 4:

Section 1

The LS-BP algorithm is based on a least square method and a BP-neural network. The LS-BP algorithm is based on a least square method and a BP-neural network. This algorithm firstly uses the least squares method to obtain prediction residuals, then trains a BP-neural network to obtain a compensation function, and finally obtains a prediction function with smaller prediction errors.

Section 2.5

The signals of the dual-channel signal reading were the absorbances of color development solution of the DPD colorimetric method and the OTO colorimetric method. The principles of these two colorimetric methods are shown in Figure.5 [51]. The principle of the DPD colorimetric method is that N, N-diethyl-p-phenylenediamine reacts with residual chlorine under acidic conditions, and forms a red compound. The principle of the OTO colorimetric method is based on the redox reaction between o-toluidine and residual chlorine to form yellow dihydrochloric acid quinone o-toluidine, and the color reaction is yellow. The color result of the DPD colorimetric method is red, and the color result of the OTO colorimetric method is yellow. The red solution has the maximum absorbance at 490-510 nm wavelength, and the yellow solution has the maximum absorbance at 440-460 nm wavelength [52]. Light at 440-460 nm and 490-510 nm wavelengths was used to irradiate the solutions after color development, respectively.

(a) The chemical reaction equation of the DPD colorimetric method principle

(b) The chemical reaction equation of the OTO colorimetric method principle

Figure 5. The principles of these two colorimetric methods

The detection of residual chlorine in water through the DPD colorimetric method or the OTO colorimetric method may be affected by impurities in the solution. The impurities probably have a strong absorption capacity for the light of certain wavelengths. This may cause fluctuations in the absorbance of the solution, and lead to inaccurate detection results and deviation of the detection results.

When the DPD colorimetric method and the OTO colorimetric method are used simultaneously to detect residual chlorine, the wavelength difference between these two detection methods is significant, so the impurities in the solution are difficult to have a greater impact on both detection methods. Hence, dual-channel signal reading has better robustness, accuracy and smaller error than single-channel signal reading method.

The color development results of the DPD colorimetric method and the OTO colorimetric method in microfluidic chip under 490-510 nm and 440-460 nm light irradiation are shown in Figure 16, respectively.

(a) Color development of the DPD colorimetric method in microfluidic chip under 490-510 nm light irradiation

(b) Color development of the OTO colorimetric method in microfluidic chip under 440-460 nm light irradiation

Figure 6. Color development of these two detection methods in microfluidic chip

Comments 5: Result and discussion, Lines 372-402: The descriptions of the channel amplitude, channel width, and channel angular frequency are highly repetitive, which could be further integrated. Besides, the author mentioned that when investigating one factor, the other two factors remain constant. What’s the exact value? How they influence each other? Why the best channel amplitude A is 2 mm, the channel width α is 0.5 mm, and the channel angular frequency ω is 2 rad s-1 in the single factor analysis, while the comprehensive analysis obtained inconsistent results (1.8 mm, 0.7mm, and 0.7 rad s-1, respectively)?

Respnse 5:

The authors sincerely thank you for your insightful comments on the Result and Discussion section of the manuscript.

  • Regarding the repetitive descriptions of channel amplitude, channel width, and channel angular frequency, the authors acknowledge the need for better integration to enhance clarity and reduce redundancy. In the revised manuscript, they will streamline these descriptions by consolidating the information where appropriate, ensuring a more concise presentation of their findings.
  • Regarding your query about the exact values and their relationships in the experiments: when investigating one factor, the other two factors were kept constant at specific values. The authors will clarify these exact values in the revised manuscript and provide a more thorough explanation of their interrelationships.
  • The best channel amplitude Aof 2 mm, channel width α of 0.5 mm, and channel angular frequency ω of 2 rad·s-1 identified in the single factor analysis were determined based on the isolated effects of each parameter on the system's performance. In this analysis, each factor was varied independently to identify its individual optimal value, which can sometimes lead to results that do not account for interactions between parameters.

In contrast, the comprehensive analysis, which yielded values of 1.8 mm for A, 0.7 mm for α, and 0.7 rad·s-1 for ω, considers the interactions among multiple factors simultaneously. This approach often leads to different optimal values as it reflects a more holistic view of the system behavior, accounting for the combined effects of the parameters.

Based on the above analysis, authors have drawn figures of the changes in the three factors. By analyzing the mixing efficiency of different cross-sections, it can obtain the comprehensive impact of the three factors on the mixing efficiency. And to clarify these discrepancies, the authors will include a discussion in the revised manuscript that explains the rationale behind the different optimal values, emphasizing the importance of both analyses. Additionally, authors will explore how the interactions between these factors may have influenced the results in the comprehensive analysis.

These revisions will aim to improve the coherence and comprehensibility of the manuscript. Thank you for guiding the authors towards enhancing the quality of the research.

Action 5:

Section 3.1

When the channel width α is 1 mm and the channel angular frequency ω is 1 rad·s-1, the liquid mixing efficiency increases accordingly as the channel amplitude A increases from 0.5 mm to 2 mm.

When the channel amplitude A is 1 mm and the channel angular frequency ω is 1 rad·s-1, the liquid mixing efficiency decreases accordingly as the channel width α increases from 0.5 mm to 2 mm.

When the channel amplitude A is 1 mm and the channel width α is 1 mm, the liquid mixing efficiency increases accordingly as the channel angular frequency ω increases from 0.5 rad·s-1 to 2 rad·s-1.

In the case of analyzing three factors separately, when the channel amplitude A is 2 mm, the channel width α is 0.5 mm, and the channel angular frequency ω is 2 rad·s-1, the liquid mixing efficiency is relatively high, which is beneficial for liquid mixing in microfluidic chips. When the channel amplitude and channel angular frequency increase, the degree of channel bending increases separately, and the Reynolds number increases. Consequently, as the Reynolds number increases, the mixing efficiency also improves. When the channel width increases, the channel widens, the liquid flow rate decreases, the Reynolds number decreases, and the mixing efficiency decreases. This result indicates that within a certain range, an increase in channel amplitude and channel angular frequency has a positive effect on liquid mixing in microfluidic chips, while an increase in channel width has a negative effect.

(a)    Mixing efficiency at 5 mm cross-section under different values of each factor

(b)   Mixing efficiency at 10 mm cross-section under different values of each factor

(c)    Mixing efficiency at 15 mm cross-section under different values of each factor

(d)   Mixing efficiency at outlet cross-section under different values of each factor

Figure11. The influence of three factors on liquid mixing efficiency.

Comments 6: Conclusion: The limitations of the present study should be addressed.

Respnse 6: The authors thank you for your insightful feedback on the conclusion section. The authors acknowledge the need to address the limitations of the study more explicitly. In the revised manuscript, the authors will provide a detailed discussion on the limitations encountered during the research, aiming to enhance the clarity and transparency of our findings.

Action 2: The following content was added to the manuscript.

Section 3.3

The portable residual chlorine detection device can achieve accurate, rapid, low-cost and convenient detection of the residual chlorine in water, filling the gap in residual chlorine detection field, but there are still some unresolved issues. There are some difficulties in mass production of the portable residual chlorine detection device. Each sensor used may have certain deviations and needs to be calibrated before use, otherwise it will cause significant deviations in the detection results. Calibration will take a lot of time and manpower. There is still room for optimization in the volume and quality of the detection device. The surface quality of microfluidic chips prepared using a photopolymerization 3D printer cannot be guaranteed, therefore further exploration of preparation factors is needed. Therefore, in future research, efforts will be made to address these issues.

Round 2

Reviewer 2 Report

Comments and Suggestions for Authors

All concerns have been addressed.